# Reverse Diffusion Sequential Monte Carlo Samplers

**Luhuan Wu**[*]
Columbia University

**Yi Han**
Columbia University

**Christian A. Naesseth**
University of Amsterdam

**John P. Cunningham**
Columbia University

## Abstract

We propose a novel sequential Monte Carlo (SMC) method for sampling from unnormalized target distributions based on a reverse denoising diffusion process. While recent diffusion-based samplers simulate the reverse diffusion using approximate score functions, they can suffer from accumulating errors due to time discretization and imperfect score estimation. In this work, we introduce a principled SMC framework that formalizes diffusion-based samplers as proposals while systematically correcting for their biases. The core idea is to construct informative intermediate target distributions that progressively steer the sampling trajectory toward the final target distribution. Although ideal intermediate targets are intractable, we develop *exact approximations* using quantities from the score estimation-based proposal, without requiring additional model training or inference overhead. The resulting sampler, termed *Reverse Diffusion Sequential Monte Carlo*, enables consistent sampling and unbiased estimation of the target's normalization constant under mild conditions. We demonstrate the effectiveness of our method on a range of synthetic targets and real-world Bayesian inference problems. [2]

## 1 Introduction

Sampling from unnormalized target distributions is a fundamental problem in many applications, ranging from Bayesian inference [1, 2] to simulating molecular systems [3]. Classical methods like Markov chain Monte Carlo (MCMC) simulate a Markov chain with the target as its stationary distribution, but they can suffer from slow mixing and difficulty traversing between modes for complex distributions. Particle methods such as importance sampling generate exact samples in a large compute limit; yet they struggle with the curse of dimensionality [4]. Alternatively, variational inference (VI) [5] casts inference as an optimization task, though its success depends on the expressiveness of the variational family and the complexity of the optimization landscape.

Recently, diffusion models have emerged as a powerful approach for sampling from complex distributions [6, 7]. They define a forward noising process that gradually transforms a complex target distribution into a simple base distribution. A reverse denoising process then reconstructs target samples by simulating the dynamics backward in time, starting from the base distribution and guided by a time-dependent score function. In the generative modeling setting, this score function is approximated by a neural network trained on samples from the target distribution. However, in the sampling context, such training data are unavailable and only an unnormalized target density is accessible.

Recent works on diffusion-based samplers explore alternative ways to approximate the score function directly from the target density, enabling sampling without access to training data. One line of works,

---

[*]Correspondence email: `lw2827@columbia.edu`

[2]Our code is available at `https://github.com/LuhuanWu/RDSMC`.

known as *diffusion Monte Carlo (MC) samplers*, estimates the score function using MC methods. Huang et al. [8], Grenioux et al. [9] consider Langevin-style MC algorithms, but they rely on a good initialization of the reverse diffusion process to ensure theoretical guarantees. He et al. [10] propose an alternative scheme based on rejection sampling which relaxes prior assumptions and improves sampling efficiency in low-dimensional regimes. These approaches demonstrate both theoretical and empirical advantages over conventional MCMC methods, particularly for multi-modal distributions.

In contrast to relying on MC estimation during sampling time, a complementary line of research trains a neural network in advance to approximate the score function. To this end, some works propose new score matching objectives [11, 12, 13], while variational approaches instead optimize divergences between forward and reverse diffusion processes [14, 15, 16, 17, 18].

While promising, diffusion-based samplers suffer from two sources of bias: discretization error in simulating the reverse diffusion process and approximation error in the estimated or learned score function. An exception is Phillips et al. [11] which mitigates the bias of a *trained* diffusion-based sampler using Sequential Monte Carlo (SMC), a general inference tool for sequential models [19, 20]. However, such training-based methods remain computationally complex compared to classical sampling methods and often rely on special neural network preconditioning [21].

To address these challenges, we develop a new diffusion-based sampler, *Reverse Diffusion Sequential Monte Carlo* (RDSMC) for sampling from unnormalized target distributions, which is training-free and admits theoretical guarantees. Inspired by prior work, we formalize diffusion MC samplers as proposal mechanisms within an SMC framework. At a high level, RDSMC generates multiple *particles* from the reverse diffusion dynamics using MC-based score estimates. To correct the bias in proposals, we introduce intermediate target distributions that guide resampling of particles at each step, progressively steering them toward the final target distribution of interest.

Crucially, our intermediate targets are efficiently computed using byproducts of MC-based score estimates, incurring no additional cost. Moreover, they form an *exact approximation* to the ideal intermediate targets that maximize sampling efficiency, defined by the marginal distributions of an extended final target [19, 22]. This design helps particles stay closely aligned with the final target throughout the sampling process. In contrast, Phillips et al. [11] learn neural network-based surrogates that require additional training and attain the ideal target only at the final step.

RDSMC belongs to a class of nested SMC methods [23, 24], inheriting the standard SMC guarantees. In particular, it produces asymptotically exact samples from the target in the limit of many particles, and provides an unbiased estimate of the normalization constant for any fixed size of particles.

Our contributions are summarized as follows:

- We propose a new SMC algorithm, Reverse Diffusion Sequential Monte Carlo (RDSMC), based on the reverse diffusion process for sampling from unnormalized distributions.
- RDSMC is training-free and extends existing diffusion MC samplers to achieve asymptotically exact sampling and provide unbiased estimates of the normalization constant, while incurring almost no computational overhead given the same number of final samples.
- Empirically, RDSMC outperforms or matches existing diffusion MC samplers and classical geometric annealing-based Annealed Importance Sampling (AIS) [25] and SMC samplers [26] on synthetic targets and Bayesian logistic regression benchmarks.

## 2  Background

**Diffusion models.**  Diffusion models [6, 7] evolve a complex target distribution $\pi(x)$ into a simple base distribution $\pi_1(x)$, e.g. $\pi_1(x) = \mathcal{N}(0, 1)$, via a forward stochastic differential equation (SDE)

$$dx_t = f(t)x_t dt + g(t)dB_t, \qquad t : 0 \to 1, \tag{1}$$

where $f(t)$ and $g(t)$ are the drift and diffusion coefficients, and $B_t$ is the standard Brownian motion.

To generate samples from $\pi(x)$, we simulate a reverse SDE initialized from $x_1 \sim \pi_1(x)$,

$$dx_t = \left[ f(t)x_t - g(t)^2 \nabla_{x_t} \log \pi_t(x_t) \right] dt + g(t)d\bar{B}_t, \qquad t : 1 \to 0, \tag{2}$$

where $\nabla_{x_t} \log \pi_t(x_t)$ is the score function of the marginal density $\pi_t(x_t)$ at time $t$ induced by the forward process in Eq. (1), and $\bar{B}_t$ is the reverse-time Brownian motion.

**Diffusion MC samplers.** Building on the diffusion model paradigm, recent works use MC methods to simulate the reverse dynamics in Eq. (2) whose terminal distribution corresponds to the target distribution $\pi \propto \tilde{\pi}$ of interest [8, 9, 10]. While the score function $\nabla_{x_t} \log \pi_t(x_t)$ is generally intractable, the key idea is to construct MC estimates via the denoising score identity [DSI, 27, 28],

$$\nabla_{x_t} \log \pi_t(x_t) = \int \frac{\alpha(t)x_0 - x_t}{\sigma(t)^2} \pi(x_0 \mid x_t) dx_0, \tag{3}$$

where $\pi(x_0 \mid x_t) \propto \tilde{\pi}(x_0)\mathcal{N}(x_0 \mid \alpha(t)x_0, \sigma(t)^2\mathbb{I})$ is the *denoising posterior*, which treats the unnormalized target $\tilde{\pi}(x_0)$ as the prior and the forward transition density $\mathcal{N}(x_0 \mid \alpha(t)x_0, \sigma(t)^2\mathbb{I})$ from Eq. (1) as the likelihood. The coefficients $\alpha(t)$ and $\sigma(t)$ are determined by the drift and diffusion terms $f(t)$ and $g(t)$ (see Appendix A.1 for details).

Eq. (3) suggests that score estimation can be cast as a posterior inference problem. In practice, one can draw approximate samples from the denoising posterior to form an MC estimate of the score, which is then substituted into the reverse dynamics to generate samples from $\pi$. Other score identities beyond the DSI can also be leveraged (see Appendix A.2).

**Sequential Monte Carlo.** SMC [19, 20] is a particle-based method for sampling from a sequence of distributions defined on variables $x_{0:T}$, terminating at a final target distribution of interest. We consider a reverse-time formulation, where SMC evolves a weighted collection of $N$ particles $\{x_t^{(i)}, w_t^{(i)}\}_{i=1}^N$ from $t = T$ to $0$, gradually approximating the final target.

An SMC sampler requires two key design choices [19], a sequence of *intermediate proposals*, $q_T(x_T)$ and $\{q_t(x_t \mid x_{t+1})\}_{t=0}^{T-1}$, and a sequence of (unnormalized) *intermediate target distributions*, $\{\gamma_t(x_{t:T})\}_{t=0}^T$, such that the final target $\gamma_0(x_{0:T})$ recovers the distribution of interest.

SMC initializes $N$ particles $x_T^{(i)} \sim q_T(x_T)$ with weights $w_T^{(i)} \leftarrow \gamma_T(x_T^{(i)})/q_T(x_T^{(i)})$ for $i = 1, \cdots, N$. Then for each step $t = T-1, \ldots, 0$ and particle $i = 1, \ldots, N$, it proceeds as follows:

1. resample ancestor $x_{t+1}^{(i)} \sim \text{Multinomial}(x_{t+1}^{(1:N)}, w_{t+1}^{(1:N)})$;

2. propagate particle $x_t^{(i)} \sim q_t(x_t \mid x_{t+1}^{(i)})$;

3. compute weight $w_t^{(i)} \leftarrow \gamma_t(x_{t:T}^{(i)}) / \left[\gamma_{+1}(x_{t+1:T}^{(i)})q_t(x_t^{(i)} \mid x_{t+1}^{(i)})\right]$.

The final set of weighted particles forms a discrete approximation to the final target $\gamma_0$, which is asymptotically exact given infinite particles under regularity conditions [20]. However, the efficacy of SMC greatly depends on the choice of intermediate targets and proposals – the closer these intermediate distributions are to the marginals of the final target, the more effective the SMC sampler.

## 3 Method

Our goal is to sample from a target distribution $\pi(x) = \frac{1}{Z}\tilde{\pi}(x)$, where $\tilde{\pi}(x)$ is the unnormalized target and $Z = \int \tilde{\pi}(x)\mathrm{d}x$ is a generally intractable normalization constant. We develop Reverse Diffusion Sequential Monte Carlo (RDSMC), a diffusion-based SMC sampler targeting $\pi(x)$.

To enable sequential inference with SMC, we define an *extended target distribution* over a discretized trajectory $x_{0:T} = (x_0, \ldots, x_T)$ of the forward process in Eq. (1),

$$\pi(x_{0:T}) = \pi(x_0) \prod_{t=1}^T \pi(x_t|x_{t-1}), \tag{4}$$

where $0 = \tau_0 < \cdots < \tau_T = 1$ are $T+1$ discretization times and $\pi(x_t|x_{t-1})$ is the forward transition density from time $\tau_{t-1}$ to $\tau_t$ induced by Eq. (1) (see Appendix A.1 for analytical expressions). Without loss of generality, we assume a uniform discretization step size $\delta = 1/T = \tau_t - \tau_{t-1}, \forall t = 1, \cdots, T$.

This construction yields a sequence of intermediate targets $\{\pi(x_{t:T})\}_{t=T}^0$, whose final marginal $\pi(x_0)$ recovers the desired target. These intermediate targets provide a natural setting for SMC, and in fact, are *optimal* intermediate gargets. However, they are intractable to sample from and to evaluate.

To address this challenge, we develop RDSMC leveraging the dual structure of diffusion processes, generating samples in the reverse direction while grounding their targets in the forward direction. In § 3.1, we introduce a sequence of proposals based on the reverse diffusion process using MC score estimates. To correct the resulting proposal bias, in § 3.2, we develop practical intermediate targets that form *exact approximations* to the optimal targets $\{\pi(x_{t:T})\}_{t=0}^{T}$ using byproducts of MC score estimates. Finally in § 3.3, we present the full RDSMC algorithm and its theoretical guarantees.

**Notation.** We slightly abuse notation by letting $x_t$ denote both the continuous-time variable $x_t$ for $t \in [0, 1]$, as used in Eqs. (1) and (2), and the discrete-time variable for $t \in \{0, \ldots, T\}$, as used throughout this section. We denote $f_t := f(\tau_t), g_t := g(\tau_t), \alpha_t := \alpha(\tau_t), \sigma_t := \sigma(\tau_t)$ for time-dependent diffusion coefficients from Eqs. (1) and (3). For generality, we define $u_t$ as the collection of *all* auxiliary random variables generated in the MC score estimation at step $t$.

## 3.1 Reverse diffusion proposal

We design an extended proposal distribution based on the reverse diffusion process with MC score estimates, jointly modeling diffusion dynamics $x_t$ and auxiliary randomness $u_t$ from score estimation.

**MC score estimation.** Given a sample $x_t$ at step $t$, we define a generic MC score estimator $s(x_t, u_t) \approx \nabla_{x_t} \log \pi(x_t)$ based on the DSI in Eq. (3), where $u_t$ is the estimation randomness associated with some sampling distribution $q(u_t \mid x_t)$.

As an illustrative example, consider an importance sampling (IS) estimator (Algorithm 2) targeting the posterior $\pi(x_0 \mid x_t) \propto \tilde{\pi}(x_0) \mathcal{N}(x_t \mid \alpha_t x_0, \sigma_t^2 \mathbb{I})$. We draw $M$ importance samples $u_t^{(m)}$ from the proposal $q(u_t^{(m)} \mid x_t) := \mathcal{N}(u_t^{(m)} \mid x_t/\alpha_t, \sigma_t^2/\alpha_t^2 \mathbb{I})$ for $m = 1, \cdots, M$, and estimate the score by

$$\nabla_{x_t} \log \pi(x_t) \approx s(x_t, u_t) := \sum_{m=1}^{M} \frac{w^{(m)}}{\sum_{m'=1}^{M} w^{(m')}} \cdot \frac{\alpha_t u_t^{(m)} - x_t}{\sigma_t^2}, \tag{5}$$

which corresponds to an MC approximation of the RHS of Eq. (3). The importance weights $w^{(m)}$ are

$$w^{(m)} = \frac{\tilde{\pi}(u_t^{(m)}) \mathcal{N}(x_t \mid \alpha_t u_t^{(m)}, \sigma_t^2 \mathbb{I})}{q(u_t^{(m)} \mid x_t)}, \qquad m = 1, \cdots, M. \tag{6}$$

In this case, the auxiliary randomness is to the collection of all importance samples $u_t = \{u_t^{(m)}\}_{m=1}^{M}$.

In practice, we use a more sophisticated AIS scheme to improve the accuracy of score estimates (Algorithm 3). More informative IS proposals or other MC methods such as SMC and rejection sampling may also be used; see Appendix B.1 for further details.

**Approximate reverse diffusion dynamics.** With the score estimates in place, we approximate the transition kernel of the reverse SDE in Eq. (2) to define a conditional proposal for $x_t$,

$$q(x_t \mid x_{t+1}, u_{t+1}) := \mathcal{N}(x_t \mid x_{t+1} - \left[ f_{t+1} x_{t+1} - g_{t+1}^2 s(x_{t+1}, u_{t+1}) \right] \delta, g_{t+1}^2 \delta). \tag{7}$$

**Extended proposal distributions.** The full sampling process starts by drawing $x_T$ from a tractable base distribution $q(x_T)$, e.g. $q(x_T) = \mathcal{N}(0, 1)$. We then iteratively evolve $x_t$ for $t = T - 1, \cdots, 0$ using the reverse diffusion dynamics $q(x_t \mid x_{t+1}, u_{t+1})$ in Eq. (7) where auxiliary variable $u_{t+1} \sim q(u_{t+1} \mid x_{t+1})$ is sampled in previous iteration to estimate the score $s(x_{t+1}, u_{t+1})$.

The overall sampling process defines a proposal over the extended space of $\{x_t, u_t\}_{t=0}^{T}$,

$$q(x_{0:T}, u_{0:T}) := q(x_T) q(u_T \mid x_T) \prod_{t=0}^{T-1} q(x_t \mid x_{t+1}, u_{t+1}) q(u_t \mid x_t). \tag{8}$$

While sampling $u_0$ at the final step $t = 0$ is not necessary, we retain it for notation simplicity.

This sampling procedure builds on prior work of Huang et al. [8], Grenioux et al. [9], which estimate the score function using MCMC, and by He et al. [10] which use rejection sampling. We formalize these ideas by defining an extended proposal distribution that incorporates randomness in MC score estimates. In particular, we adopt an IS or AIS-based MC approach that also provides an unbiased estimate of the normalization constant, a property that, in principle, can be achieved by the rejection sampling approach of He et al. [10] as well. This property will be leveraged in the next part to construct intermediate targets for RDSMC.

## 3.2 Intermediate targets

Samples from the reverse diffusion proposal deviate from the desired target $\pi$ due to two sources of error: time discretization of the reverse SDE, and bias in score estimation. To correct these errors, we design a series of intermediate target distributions. We first characterize the optimal but intractable targets, and then develop practical approximations using byproducts of score estimation. Finally, we extend the intermediate targets to include auxiliary variables. Our construction aligns with the optimal targets at intermediate steps and recovers the desired marginal $\pi(x_0)$ at the final step $t = 0$.

**Optimal intermediate targets.** The optimal intermediate target at step $t$ is the marginal distribution of the extended target $\pi(x_{0:T})$ in Eq. (4),

$$\pi(x_{t:T}) = \int \pi(x_{0:T}) \mathrm{d}x_{0:t-1} = \pi(x_t) \prod_{i=t+1}^{T} \pi(x_i \mid x_{i-1}), \tag{9}$$

as it leads to exact samples from $\pi(x_{0:T})$ when combined with locally optimal proposals [19].

However, the marginal $\pi(x_t) = \int \pi(x_0) \mathcal{N}(x_t \mid \alpha_t x_0, \sigma_t^2 \mathbb{I}) \mathrm{d}x_0$ in Eq. (9) involves an intractable integral, except in the final step $t = 0$, where $\pi(x_0)$ is known up to a normalization constant.

**Marginal estimation.** For $t > 0$, we approximate the marginal $\pi(x_t)$ by $\hat{\pi}(x_t, u_t)$ using byproducts of MC score estimates, where we recall $u_t$ is the estimation randomness. The key insight is that $Z\pi(x_t)$ is the normalization constant of the posterior $\pi(x_0 \mid x_t) \propto \tilde{\pi}(x_0) \mathcal{N}(x_t \mid \alpha_t x_0, \sigma_t^2 \mathbb{I})$, that is,

$$\int \tilde{\pi}(x_0) \mathcal{N}(x_t \mid \alpha_t x_0, \sigma_t^2 \mathbb{I}) \mathrm{d}x_0 = Z \int \pi(x_0) \mathcal{N}(x_t \mid \alpha_t x_0, \sigma_t^2 \mathbb{I}) \mathrm{d}x_0 = Z\pi(x_t).$$

Hence, we can re-use the posterior inference procedure in MC score estimation to obtain an unbiased estimate of the desired marginal $\pi(x_t)$ (up to a factor of $Z$).

For example, IS or AIS-based score estimation (Algorithms 2 and 3) generates importance weights $\{w^{(m)}\}_{m=1}^{M}$ targeting $\pi(x_0 \mid x_t)$ for a fixed $x_t$; we then obtain an unbiased marginal estimate

$$\hat{\pi}(x_t, u_t) := \frac{1}{M} \sum_{m=1}^{M} w^{(m)}, \text{ with } \mathbb{E}_{q(u_t \mid x_t)}[\hat{\pi}(x_t, u_t)] = Z\pi(x_t). \tag{10}$$

At the final step $t = 0$, we set $\hat{\pi}(x_0, u_0) := \tilde{\pi}(x_0) = Z\pi(x_0)$ as no approximation is needed.

**Extended intermediate targets.** Finally, we incorporate the auxiliary variables $u_t$ by setting their targets to match their sampling distributions and define the extended intermediate target as

$$\gamma_t(x_{t:T}, u_{t:T}) := \hat{\pi}(x_t, u_t) \, q(u_t \mid x_t) \prod_{i=t}^{T-1} \pi(x_{i+1} \mid x_i) \, q(u_{i+1} \mid x_{i+1}) \tag{11}$$

for $t = 0, \cdots, T-1$, and $\gamma_T(x_T, u_T) := \hat{\pi}(x_T, u_T) \, q(u_T \mid x_T)$.

The structure in Eq. (11) mirrors that of the optimal targets in Eq. (9), replacing the intractable marginal $\pi(x_t)$ with an unbiased estimate $\hat{\pi}(x_t, u_t)$ (up to a factor of $Z$) and accounting for auxiliary randomness $u_{t:T}$. Moreover, we make the following two observations.

*Observation* 1. The final marginal target at $t = 0$ matches the desired $\pi(x_0)$ with *no approximations*,

$$\begin{aligned}
\gamma_0(x_0) &= \int \gamma_0(x_{0:T}, u_{0:T}) \mathrm{d}x_{1:T}, u_{0:T} \\
&= \int \hat{\pi}(x_0, u_0) q(u_0 \mid x_0) \prod_{i=1}^{T} \pi(x_i \mid x_{i-1}) \, q(u_i \mid x_i) \mathrm{d}x_{1:T}, u_{0:T} \\
&= \tilde{\pi}(x_0) \propto \pi(x_0),
\end{aligned} \tag{12}$$

as $\hat{\pi}(x_0, u_0) = \tilde{\pi}(x_0)$ by construction. This implies our final SMC target is correctly specified.

---

**Algorithm 1:** Reverse Diffusion Sequential Monte Carlo (RDSMC)

---

**Input:** Unnormalized target $\tilde{\pi}(x_0)$, number of particles $N$, discretization steps $T$ (with step size $\delta = 1/T$), diffusion schedule $\{\alpha_t, \sigma_t, f_t, g_t\}$, base distribution $q(x_T)$, and additional inputs for score and marginal estimation $\eta$

**Output:** Weighted samples $\{x_0^{(i)}, w_0^{(i)}\}_{i=1}^N$ and normalization constant estimate $\hat{Z}$

1 **for** $i \leftarrow 1$ **to** $N$ **do**
2    Sample $x_T^{(i)} \sim q(x_T)$;
3    Compute score and marginal estimates $s_T^{(i)}, \hat{\pi}_T^{(i)} \leftarrow$ Algorithm 3$(\pi, \alpha_T, \sigma_T, x_T^{(i)}, \eta)$;
4    Compute weight: $w_T^{(i)} \leftarrow \frac{\hat{\pi}_T^{(i)}}{q(x_T^{(i)})}$;

5 **for** $t \leftarrow T - 1$ **to** $0$ **do**
6    Resample $\{x_{t+1}^{(i)}, u_{t+1}^{(i)}, s_{t+1}^{(i)}, \hat{\pi}_{t+1}^{(i)}\}_{i=1}^N$ according to weights $\{w_{t+1}^{(i)}\}_{i=1}^N$ ;
7    **for** $i \leftarrow 1$ **to** $N$ **do**
8      Sample $x_t^{(i)} \sim q(x_t \mid x_{t+1}^{(i)}, u_{t+1}^{(i)}) := \mathcal{N}\left(x_t \mid x_{t+1}^{(i)} - \left[f_{t+1}x_{t+1}^{(i)} - g_{t+1}^2 s_{t+1}^{(i)}\right]\delta, g_{t+1}^2\delta\right)$;
9      **if** $t > 0$ **then**
10        Compute score and marginal estimates $s_t^{(i)}, \hat{\pi}_t^{(i)} \leftarrow$ Algorithm 3 $(\pi, \alpha_t, \sigma_t, x_t^{(i)}, \eta)$;
11      **else**
12        Compute exact marginal $\hat{\pi}_0^{(i)} \leftarrow \tilde{\pi}(x_0^{(i)})$;
13      Compute weight: $w_t^{(i)} \leftarrow \frac{\hat{\pi}_t^{(i)} \pi(x_{t+1}^{(i)}|x_t^{(i)})}{\hat{\pi}_{t+1}^{(i)} q(x_t^{(i)}|x_{t+1}^{(i)}, u_{t+1}^{(i)})}$;

14 Compute normalization constant estimate $\hat{Z} \leftarrow \prod_{t=0}^{T} \frac{1}{N} \sum_{i=1}^N w_t^{(i)}$.

---

*Observation* 2. Marginalizing out $u_{t:T}$ in $\gamma_t(x_{t:T}, u_{t:T})$ recovers the optimal target $\pi(x_{t:T}), \forall t$,

$$
\begin{aligned}
\int \gamma_t(x_{t:T}, u_{t:T})\mathrm{d}u_{t:T} &= \int \hat{\pi}(x_t, u_t)\, q(u_t \mid x_t) \prod_{i=t}^{T-1} \pi(x_{i+1} \mid x_i)\, q(u_{i+1} \mid x_{i+1})\mathrm{d}u_{t:T} \\
&= \mathbb{E}_{q(u_t|x_t)}\left[\hat{\pi}(x_t, u_t)\right] \prod_{i=t}^{T-1} \pi(x_{i+1} \mid x_i) \qquad (13) \\
&= Z\pi(x_t) \prod_{i=t}^{T-1} \pi(x_{i+1} \mid x_i) \propto \pi(x_{t:T}),
\end{aligned}
$$

where the last equality follows from the unbiasedness of marginal estimates, as in Eq. (10). This result shows that our intermediate targets $\gamma_t(x_{t:T}, u_{t:T})$ are aligned with the optimal $\pi(x_{t:T})$, despite the approximation in $\hat{\pi}(x_t, u_t)$, a property known as the *exact approximation* [22, 19].

While the generic SMC framework remains valid under alternative choices of intermediate targets, provided that $\gamma_0(x_0) \propto \pi(x_0)$, our construction offers a balance between theoretical correctness and practical efficiency. The marginal estimates $\hat{\pi}(x_t, u_t)$ in our intermediate targets from Eq. (11) act as *twisting* or *look-ahead* functions [19, Chapter 3] that incorporate future information $\pi(x_0)$ into intermediate steps. Together, Observations 1 and 2 ensure that our intermediate targets provide effective guidance throughout the sampling process while recovering the desired target in the end.

### 3.3 RDSMC: Algorithm and theoretical guarantee

We now derive weighting functions, the final component of RDSMC, which guides the resampling of particles. We then present the complete algorithm and establish its theoretical guarantees.

**Weighting functions.** Following the standard SMC framework, we define the weighting functions on the extended space $\{x_{t:T}, u_{t:T}\}$ for $t = T-1, \cdots, 0$ as ratios of intermediate targets and proposals

$$
w_t := \frac{\gamma_t(x_{t:T}, u_{t:T})}{\gamma_{t+1}(x_{t+1:T}, u_{t+1:T})\, q(x_t \mid x_{t+1}, u_{t+1})\, q(u_t \mid x_t)}. \qquad (14)
$$

Substituting the expressions for $\gamma_t$ from Eq. (11) and canceling out common terms yields

$$w_t = \frac{\hat{\pi}(x_t, u_t)\pi(x_{t+1}|x_t)}{\hat{\pi}(x_{t+1}, u_{t+1})q(x_t \mid x_{t+1}, u_{t+1})}. \tag{15}$$

At the initial step $T$, the weight is $w_T := \frac{\gamma_T(x_T, u_T)}{q(x_T)q(u_T|x_T)} = \frac{\hat{\pi}(x_T, u_T)}{q(x_T)}$ recalling that $\gamma_T(x_T, u_T) = \hat{\pi}(x_T, u_T)q(u_T \mid x_T)$ by construction.

Notably, although the intermediate targets and proposals involve auxiliary sampling distributions $q(u_t \mid x_t)$ for score estimation, the weighting functions $w_t$ in Eq. (15) and $w_T$ do *not* depend on their evaluation. Hence, a generic MC sampler can be used while still retaining computable weights, as long as it produces tractable estimates of the score and marginal densities.

**The RDSMC algorithm and theoretical guarantees.** We summarize the RDSMC algorithm in Algorithm 1. It initializes $N$ particles from a tractable base distribution. Subsequently, the particles are propagated through a reverse diffusion-based proposal, followed by weighting and resampling using intermediate targets. These steps are enabled by running an inner-level MC estimation of the score function and the marginal density. In this work, we use AIS (Algorithm 3) while other methods can be incorporated as well (see Appendix B.1). The final output of RDSMC is a weighted set of samples approximating the target $\pi(x_0)$, along with an estimate of the normalization constant $Z$.

RDSMC can be viewed as an adaptation of the *nested* SMC framework [23, 24]. The inner-level MC estimation involves a sampling procedure (e.g. AIS) targeting the intractable posterior $\pi(x_0 \mid x_t)$, which is then used to assign *proper weights* [19, Chapter 4.3] to proposed samples $x_t$. Consequently, RDSMC inherits the unbiasedness and asymptotic exactness guarantees of nested SMC [23, 19].

**Theorem 1** (Informal). *Under regularity conditions, the RDSMC algorithm provides a consistent estimator of the target distribution $\pi(x_0)$ as particle size $N \to \infty$ and an unbiased estimator of the normalization constant $Z$ for any $N \geq 1$.*

We provide the formal statement and proof in Appendix C.

Theorem 1 shows that RDSMC effectively wraps a diffusion MC-style proposal within an SMC framework, mitigating its bias as the number of particles increases. Moreover, it provides an unbiased estimate of the normalization constant, a property absent in existing diffusion MC methods [8, 9, 10].

The bias correction mechanism of RDSMC arises from two key aspects: (1) the final target of RDSMC is explicitly constructed to match the desired target $\pi(x_0)$, regardless of the discretization scheme or marginal density approximation (Observation 1); and (2) errors in score estimation affect only the proposal steps, which are accounted for in the weighting functions. Therefore, the weighting and resampling steps in the outer SMC loop asymptotically correct for proposal bias, ensuring convergence to the final target $\pi(x_0)$ in the limit of many particles.

## 4 Related works

**Diffusion MC samplers.** Huang et al. [8] develop the RDMC sampler based on the reverse diffusion process using Langevin-style MCMC to estimate the score function. They initialize the sampler at some $\tau_T < 1$ via a nested Langevin procedure. Grenioux et al. [9] propose a similar SLIPS sampler with a signal-to-noise-ratio (SNR)-adjusted discretization scheme, and provide further guidance on choosing the initial sampling time $\tau_T$ under suitable conditions. In contrast, we use AIS-based score estimators to enable marginal estimation and start sampling from the base distribution at $\tau_T = 1$.

Alternatively, He et al. [10] explore rejection sampling for score estimation, improving the sampling efficiency in low-dimensional settings, which may be incorporated into our framework as well.

While these methods provide certain theoretical guarantees, we offer an orthogonal means to improve their accuracy by increasing the number of particles. RDSMC can be viewed as a generic SMC wrapper around these diffusion-MC samplers (given suitable score estimators), enabling consistent sampling and unbiased estimation of the normalization constant.

**Training-based diffusion samplers** In addition to developing MC estimates of the score function, another line of work trains a neural network directly using unnormalized target information.

Variational approaches achieve this goal by minimizing the divergence between forward and reverse processes [14, 15, 17, 18, 16], while Akhound-Sadegh et al. [12], OuYang et al. [13] exploit various identities of the score function or a related energy function. Similar to their training-free MC-based counterparts, these methods are prone to discretization and score approximation errors.

Closest to our work is Phillips et al. [11] that also use SMC to correct proposal bias. Their method requires training neural networks, while ours is training-free. Moreover, their neural network-based intermediate targets incur approximation errors except at the last step, while ours reflect the true target marginals at each SMC iteration. See an empirical comparison in Appendix E.5.3.

He et al. [21] show that many training-based methods incur heavy computational overhead compared to classical sampling methods and rely on special network preconditioning. For this reason, we limit our empirical comparison to training-free samplers. Nonetheless, an interesting direction for future work is to combine our method with training-based approaches to further enhance performance.

**SMC for conditional generation from diffusion models.**   Beyond classical sampling tasks, SMC has been applied to conditional generation for pre-trained diffusion models [29, 30, 31, 32]. While these works also combine SMC and diffusion dynamics, a key distinction is the target distribution: they sample from an existing diffusion model prior tilted with a reward function, whereas our setting involves sampling from any (unnormalized) target distribution. As a result, the way diffusion models are used is different, and we require distinct designs for proposals and intermediate targets in SMC.

Nevertheless, RDSMC uses the estimated marginal densities in Eq. (11) as twisting functions to improve SMC sampling efficiency. This strategy is conceptually is similar to that of Wu et al. [30], though in a different setting; see Appendix D for further discussion.

## 5   Experiments

We evaluate RDSMC on a range of synthetic and real-world target distributions, comparing it to SMC [26], AIS [25], SMS [33], RDMC [8], and SLIPS [9]. AIS and SMC operate over a series of geometric interpolations between the target and a Gaussian proposal with MCMC transitions. SMS samples from the target by sequentially denoising equally noised measurements using a Langevin procedure combined with estimated scores. RDMC and SLIPS are described in § 4.

RDSMC uses a variance-preserving diffusion schedule. To evaluate the effectiveness of our intermediate targets and the importance of resampling, we include two ablations: (i) RDSMC (Proposal), which samples directly from the reverse diffusion proposal in Eq. (8) without any weighting or resampling, and (ii) RDSMC (IS), which applies a final IS correction to samples from RDSMC (Proposal).

Baseline methods follow the implementation of Grenioux et al. [9]. Notably, the information about the target variance (or an estimation) is provided to guide the initialization of the baselines SMC, AIS, SMS and SLIPS. In contrast, our method does *not* make use of this extra information.

Unless otherwise specified, we use $T = 100$ discretization steps for RDSMC and its variants, and $T = 1,024$ steps for other methods. We generate $N = 4,096$ final samples for all methods, and tune their hyperparameters assuming access to either a validation dataset or an oracle metric.

We provide ablation studies controlling for the discretization steps, the total running time and comparable hyperparameter settings in Appendix E.5, where we observe RDSMC still remains competitive and, in some cases, superior. All experiment details are included in Appendix E.

### 5.1   Bi-modal gaussian mixtures

We first study bi-modal Gaussian mixtures with an imbalanced weight ratio of $w_1/(w_1 + w_2) = 0.1$ for varying dimensions $d$. The estimated ratio is obtained by assigning samples to the mode with the highest posterior probability. For each method we select the hyperparameters based on the lowest estimation bias of the weight ratio.

In Figure 1a, we compare the marginal histogram of samples from RDSMC, RDSMC (Proposal), and the true target in $d = 2$. While samples from RDSMC (Proposal) cover both modes, their relative weights are overly balanced. In contrast, RDSMC recovers the calibrated mode weights, indicating the importance of the error correction mechanism by SMC.

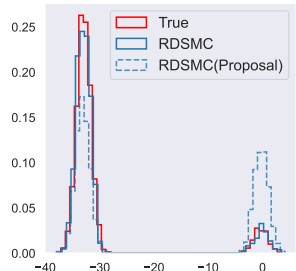 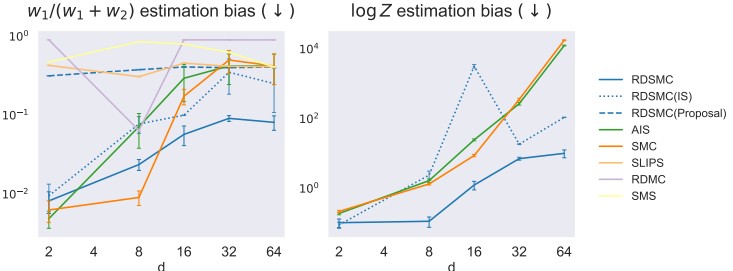

(a) $d = 2$: histogram along 1st dimension. While RDSMC's proposal covers the modes, the SMC procedure is crucial for producing calibrated weights.

(b) Estimation bias of weight ratio $w_1/(w_1 + w_2)$ and log-normalization constant $\log Z$ versus dimension $d$. Results are averaged over 5 seeds with error bars showing one standard error. Note that SMS, RDMC, and SLIPS do not provide estimate of $\log Z$. For both metrics, the bias increases with $d$ for all methods, while RDSMC outperforms the baselines for most cases.

Figure 1: Bi-modal Gaussian mixture study.

| Algorithm | Rings ($d = 2$) | | Funnel ($d = 10$) | |
|---|---|---|---|---|
| | Radius TVD ($\downarrow$) | $\log Z$ Bias ($\downarrow$) | Sliced KSD ($\downarrow$) | $\log Z$ Bias ($\downarrow$) |
| AIS | $0.10 \pm 0.00$ | $0.05 \pm 0.00$ | $0.07 \pm 0.00$ | $\mathbf{0.28 \pm 0.01}$ |
| SMC | $0.10 \pm 0.00$ | $0.05 \pm 0.00$ | $0.07 \pm 0.00$ | $\mathbf{0.28 \pm 0.01}$ |
| SMS | $0.24 \pm 0.01$ | N/A | $0.15 \pm 0.00$ | N/A |
| RDMC | $0.37 \pm 0.00$ | N/A | $0.13 \pm 0.00$ | N/A |
| SLIPS | $0.19 \pm 0.00$ | N/A | $\mathbf{0.06 \pm 0.00}$ | N/A |
| RDSMC | $0.13 \pm 0.01$ | $\mathbf{0.03 \pm 0.00}$ | $0.11 \pm 0.00$ | $\mathbf{0.28 \pm 0.10}$ |
| RDSMC(IS) | $0.15 \pm 0.01$ | $\mathbf{0.02 \pm 0.01}$ | $0.33 \pm 0.03$ | $1.61 \pm 0.14$ |
| RDSMC(Proposal) | $\mathbf{0.09 \pm 0.00}$ | N/A | $0.32 \pm 0.03$ | N/A |

Table 1: Results on Rings and Funnel (mean $\pm$ standard error over 5 seeds). Bold indicates 95% confidence interval overlap with the best average result. RDSMC has the lowest $\log Z$ estimation bias for both targets. On Rings, RDSMC(Proposal) has the lowest radius TVD, followed by AIS, SMC, and RDSMC. On Funnel, SLIPS has the lowest sliced KSD, followed by AIS, SMC, and RDSMC.

Figure 1b shows the estimation bias of the weight ratio (left), and that of the log-normalization constant $\log Z$ (right). Note that only RDSMC, RDSMC(IS), AIS and SMC provide estimates for $\log Z$. We observe that RDSMC consistently outperforms other methods in both metrics across dimensions. Moreover, RDSMC (Proposal) exhibits consistently high weight ratio estimation bias. While RDSMC (IS) reduces this bias, it still underperforms the full RDSMC procedure, highlighting the effectiveness of intermediate resampling guided by our intermediate target distributions.

In high dimensions, all methods exhibit some degree of mode collapse. RDSMC primarily samples from the dominant mode, resulting in average weight ratio estimation biases of 0.09 and 0.08 for $d = 32$ and 64, respectively. In contrast, AIS and SMC completely collapse to a single mode, which varies across runs, yielding degenerate estimated ratios of 0 or 1 and an average bias of around 0.4.

## 5.2 Rings and Funnel distributions

We present results on two additional synthetic targets. Rings, introduced by Grenioux et al. [9], is a 2-dimensional distribution constructed via an inverse polar parameterization: the angular component is uniformly distributed, while the radius component follows a 4-mode Gaussian mixture. Funnel [34], is a 10-dimensional "funnel"-shaped distribution.

For Rings, we assess sample quality using total variational distance in the radius component (Radius TVD). For Funnel, use sliced Kolmogorov-Smirnov distance (Sliced KSD) to evaluate sample quality following Grenioux et al. [9]. As both targets admit tractable normalization constants, we also report the estimation bias for methods that compute them. For each method, we select the hyperparameters with lowest Raidus TVD on a heldout validation set for Rings and lowest Sliced KSD for Funnel.

| Test LL ($\uparrow$) | Credit ($d=25$) | Cancer ($d=31$) | Ionosphere ($d=35$) | Sonar ($d=61$) |
|---|---|---|---|---|
| AIS | $\mathbf{-122.73 \pm 0.51}$ | $-60.45 \pm 0.31$ | $\mathbf{-86.37 \pm 0.10}$ | $-110.11 \pm 0.06$ |
| SMC | $-123.17 \pm 0.05$ | $-60.28 \pm 0.11$ | $\mathbf{-86.37 \pm 0.10}$ | $-110.11 \pm 0.06$ |
| SMS | $-527.79 \pm 0.85$ | $-215.64 \pm 0.66$ | $-202.56 \pm 0.16$ | $-275.44 \pm 0.31$ |
| RDMC | $-388.24 \pm 1.75$ | $-182.81 \pm 0.43$ | $-108.67 \pm 0.09$ | $-128.29 \pm 0.03$ |
| SLIPS | $\mathbf{-121.79 \pm 0.04}$ | $\mathbf{-56.26 \pm 0.08}$ | $\mathbf{-85.07 \pm 0.07}$ | $\mathbf{-102.39 \pm 0.03}$ |
| RDSMC | $\mathbf{-124.00 \pm 1.96}$ | $-62.23 \pm 1.96$ | $\mathbf{-87.72 \pm 1.75}$ | $\mathbf{-101.52 \pm 1.84}$ |
| RDSMC(IS) | $-144.38 \pm 4.63$ | $-82.47 \pm 7.78$ | $\mathbf{-84.90 \pm 2.84}$ | $-110.57 \pm 4.54$ |
| RDSMC(Proposal) | $-606.03 \pm 1.26$ | $-246.86 \pm 0.97$ | $-92.62 \pm 0.04$ | $-134.72 \pm 0.13$ |

Table 2: Bayesian logistic regression with test log-likelihood (mean $\pm$ standard error) averaged over 5 seeds. Bold indicates 95% confidence interval overlap with that of the best average result. SLIPS achieves the best overall performance, while RDSMC matches or closely approaches it. In contrast, RDSMC(Proposal) performs noticeably worse, highlighting the effectiveness of the SMC correction.

As shown in Table 1, RDSMC achieves lower or comparable estimation bias of the log-normalization constant relative to AIS and SMC on both targets. On the lower-dimensional Rings target, RDSMC (Proposal) has the lowest radius TVD, where AIS, SMC and RDSMC closely match. On the more challenging Funnel target, SLIPS achieves the lowest sliced KSD; while RDSMC performs slightly worse, it greatly outperforms its IS and Proposal variants.

### 5.3 Bayesian logistic regression

Finally, we evaluate inference performance on Bayesian logistic regression models using four datasets. Credit and Cancer [35] involve predicting credit risk and breast cancer recurrence, while Ionosphere [36] and Sonar [37] focus on classifying radar and sonar signals, respectively. The inference is performed on $60\%$ of each dataset, leaving 20% for validation and 20% for testing. Each method is tuned using the validation log-likelihood estimate. We report the test log-likelihood in Table 2.

RDSMC achieves the best or near-best performance across datasets, closely matching SLIPS, which performs the best overall. This comparison suggests that SLIPS's time discretization and initialization strategy may offer complementary benefits despite lacking systematic error correction. RDSMC (IS) underperforms on Credit and Cancer and exhibits higher variance than RDSMC, while RDSMC (Proposal) performs the worst, again highlighting the importance of the SMC mechanism. Compared to baseline methods, RDSMC shows greater variance, likely due to auxiliary randomness in its nested structure, suggesting a direction for future improvement.

## 6 Discussion

This paper presents a training-free and diffusion-based SMC sampler, Reverse Diffusion Sequential Monte Carlo (RDSMC), for sampling from unnormalized distributions. By leveraging reverse diffusion dynamics as proposals, we devise informative intermediate targets to correct systematic errors within a principled SMC framework. These components rely on Monte Carlo score estimation, without requiring additional training. Our algorithm provides asymptotically exact samples from the target distribution and a finite-sample unbiased estimate of the normalization constant. Empirical results on both synthetic and Bayesian inference tasks demonstrate competitive or superior performance compared to existing diffusion-based and classical geometric annealing-based samplers.

**Limitations and future work.** As a nested SMC procedure, RDSMC introduces auxiliary variance. Such variance may be mitigated by enhancing autocorrelation within the MC estimation step [e.g. 38]. In high-dimensional Gaussian mixture experiments, we observe a degree of mode collapse, an issue also seen in other samplers. Potential remedies include partial resampling [39], or using more informed proposals. Our method also relies on oracle metrics for hyperparameter tuning, which aligns with existing works; however, developing a more automated strategy remains an important future direction. Finally, to further improve performance, we can incorporate SNR-adapted discretization schemes and informative initializations, as explored by Grenioux et al. [9], as well as combine our approach with training-based diffusion samplers.

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

| Test LPPD ↑ | Credit ($d = 25$) | Cancer ($d = 31$) | Ionosphere ($d = 35$) | Sonar ($d = 61$) |
|---|---|---|---|---|
| RDSMC | **-94.54 ± 1.34** | **-10.59 ± 0.45** | **-25.97 ± 0.73** | **-18.54 ± 0.28** |
| RDSMC(Proposal) | -94.62 ± 0.06 | -50.24 ± 0.16 | **-24.87 ± 0.02*** | -18.91 ± 0.01 |
| RDMC | -138.22 ± 0.35 | -78.03 ± 0.12 | -44.16 ± 0.07 | -28.64 ± 0.03 |
| SLIPS | **-92.42 ± 0.02*** | -10.41 ± 0.01 | -25.22 ± 0.01 | **-18.40 ± 0.01*** |
| AIS | **-92.91 ± 0.44** | **-10.13 ± 0.07** | -25.09 ± 0.02 | **-18.41 ± 0.01** |
| SMC | **-92.55 ± 0.05** | **-10.13 ± 0.01*** | -25.09 ± 0.02 | **-18.41 ± 0.01** |
| SMS | -98.00 ± 0.27 | -20.47 ± 0.16 | -26.17 ± 0.10 | -23.29 ± 0.08 |

Table 8: Ablation experiments: Bayesian logistic regression with test log pointwise predictive density (Test LPPD, with mean ± standard error) averaged over 10 seeds. Bold indicates 95% confidence interval overlap with that of the best average result. RDSMC, AIS, SMC, and SLIPS achieve the best overall performance. RDSMC outperforms RDSMC (Proposal) and RDMC in most cases, highlighting the effectiveness of the SMC correction.

where the likelihood $p(y_i \mid x_i; w, b)$ is defined in Eq. (43) and the posterior $p(w, b \mid \mathcal{D})$ is defined in Eq. (44).

We use final (weighted) samples from each method to approximate the Test LPPD in Eq. (46).

We report the results in Table 8. We observe that RDSMC, AIS, SMC, and SLIPS achieve the best overall performance. Compared to SLIPS, RDSMC does not require target-variance-based initialization. Moreover, RDSMC outperforms RDSMC (Proposal) and RDMC in most cases, highlighting the effectiveness of the SMC correction. However, we observe that results for RDSMC exhibit large variance, as the case in the main experiments from Table 2.

The optimal hyperparameters for each method are selected based on the highest LPPD on a heldout validation set, computed analogously to the Test LPPD.

### E.5.3 Comparison to Particle Denoising Diffusion Sampler on the Funnel target

Finally, we present a comparison to the Particle Denoising Diffusion Sampler (PDDS) [11], which is also a diffusion-based SMC sampler. However, unlike RDSMC, PDDS requires training.

We evaluate PDDS on the Funnel target using 10 random seeds. Experiments are conducted on an NVIDIA RTX A6000 GPU, whereas our previous experiments use an NVIDIA A100 GPU. In addition, PDDS is implemented in JAX (following the official implementation at `https://github.com/angusphillips/particle_denoising_diffusion_sampler#` ), while RDSMC and the other baselines are implemented in PyTorch.

PDDS requires approximately $82.63 \pm 0.43$ seconds for training and $15.08 \pm 0.073$ seconds for final sampling of 4096 particles, resulting in a total runtime of about $97.70 \pm 0.47$ seconds (reported as mean ± standard error). The runtimes of our method and other training-free baselines are reported in Figure 3, which are all below 25 seconds. However, due to differences in hardware and software frameworks, PDDS's runtime is not directly comparable to those of the previous experiments.

PDDS achieves a $\log Z$ bias of $0.26 \pm 0.04$ and a mean Sliced KSD of $0.10 \pm 0.00$. Compared with the results in Table 7, PDDS exhibits a slightly lower average $\log Z$ bias than RDSMC, but higher than AIS and SMC. Its Sliced KSD is also higher than that of RDSMC, AIS, and SMC.

