# OpenReview forum: "Reverse Diffusion Sequential Monte Carlo Samplers"
_NeurIPS.cc/2025/Conference — NeurIPS 2025 poster_

### Official Review · Reviewer_8TQX · 2025-06-20

**Clarity:** 1
**Significance:** 3
**Originality:** 2
**Rating:** 4
**Confidence:** 3

**Summary:**

This paper proposes a novel SMC method to sample from unnormalized target distributions. Utilizing the diffusion framework and building upon diffusion-based SMC methods, this paper designs informative intermediate targets to guide the resampling of the particles at each step, and the new sampler is called RDSMC. Asymptotic consistency to estimate the target distribution is established theoretically as the number of particles diverge. Numerical experiments are performed on synthetic and real-world data.

**Questions:**

Some questions have been provided in the weakness section. My other questions are as follows.

1. It is claimed that the density $q(u_t|x_t)$ need not be evaluated (line 137). However, that seems to be needed to compute the importance weights (in Alg. 2). Am I missing something?

2. While in a different context, how is the design of the weighting functions similar to TDS?

I would consider raising my score if my questions are properly addressed.

**Ethical Concerns:**

["NO or VERY MINOR ethics concerns only"]

**Final Justification:**

The paper is not clear enough and needs calibration. That said, through the discussion with the authors I agree with some merit in the paper's contribution, which I think is worth being published.

**Limitations:**

Yes

**Paper Formatting Concerns:**

No major formatting concerns.

**Quality:**

2

**Strengths And Weaknesses:**

Strengths:
1. A new SMC algorithm, RDSMC, for sampling from unnormalized targets
2. RDSMC enables estimation of the normalization constant
3. The paper provides a theoretical consistency result for estimating the target and the normalization constant

Weaknesses:
1. It seems that in most reported experiments (especially with higher dimensional data), SLIPS (sometimes along with other baseline methods) is performing better than the proposed RDSMC.
2. Also, especially for estimating the normalization constant, there are some other recent diffusion-based samplers that are not included for numerical comparison. For example, while not using information from the unnormalized densities, PDDS also estimates the normalization constant which should be numerically comparable.
3. Overall, the benefits of using SMC against MCMC (e.g., RDMC) with diffusion-based sampling methods are still not clear to me. I am thinking of benefits other than the final distributional distance (e.g., computation, what is required for estimation, etc.).
4. There are occasional typos that need to be fixed: "an exception" -> "exceptions" (line 43); comma -> multiplication (line 99)

---

> ### Author Rebuttal · Authors · 2025-07-29
>
> We thank the reviewer for their thoughtful evaluation and are pleased to see they recognize the our work as a new SMC algorithm that provides unbiased normalization constant along with theoretical consistency guarantees. We also found the questions and suggestions raised helpful. We address these points below.
>
> Note: _if Latex equations do not render, please consider refreshing the page a couple of times_.
>
>
> > 1. It seems that in most reported experiments (especially with higher dimensional data), SLIPS (sometimes along with other baseline methods) is performing better than the proposed RDSMC.
>
> Respectfully, we would like to highlight that RDSMC offers strong performance across a variety of settings. Below we clarify specific points and provide supporting ablation results:
>
> - **RDSMC shows better or competitive performance without access to target variance and with fewer discretization steps:** As mentioned in line 249-255, competing methods including **AIS, SMC and SLIPS have access to the target’s scalar variance and use substantially more discretization steps** (T=1,024); SLIPS also uses an SNR-adapted time discretization schedule and a Langevin-based initialization. In contrast, RDSMC is run with T=100 uniform steps, does not use target variance and starts from a simple Gaussian $N(0,1)$. Despite these differences, RDSMC outperforms or matches these methods in most cases. In particular, RDSMC outperforms SLIPS along other baselines in the GMM weight ratio estimation task across varying dimensions $d \in [2,64]$ (Figure 2 in appendix).
>
> - As a more neutral comparison, **RDMC is similar to SLIPS but  does not assume target variance, and RDSMC consistently outperforms RDMC across multiple targets.**
>
> - **RDSMC outperforms baselines given target variance**: To further isolate the effect of our core contributions, we include an ablation experiment on rings and funnel targets (see table below), where all methods use the target scalar variance (RDSMC uses it to initialize the proposal for score estimation). In these settings, **RDSMC outperforms SMC and SLIPS on most metrics.**
>
>
> | Method                            | Rings TVD               | Rings Log Z bias        | Funnel KS               | Funnel Log Z bias        |
> |----------------------------------|--------------------------|--------------------------|--------------------------|---------------------------|
> | RDSMC (T=200, use target var)    | $\mathbf{0.102 \pm 0.002}$ | $\mathbf{0.012 \pm 0.004}$ | $\mathbf{0.065 \pm 0.002}$ | $\mathbf{0.219 \pm 0.025}$ |
> | SMC (T=1024, use target var)     | $\mathbf{0.096 \pm 0.002}$ | $0.053 \pm 0.002$         | $0.071 \pm 0.000$         | $0.283 \pm 0.005$          |
> | SLIPS (T=1024, use target var)   | $0.189 \pm 0.003$         | NA                       | $\mathbf{0.063 \pm 0.001}$ | NA                        |
>
>
> We will revise the manuscript to highlight these distinctions in settings and include ablation study results.
>
> > 2. There are other recent diffusion-based samplers that are not included for numerical comparison, e.g. PDDS.
>
> We thank the reviewer for this suggestion. As noted in the Related Works section, **PDDS requires neural network training which introduces significant computation overhead**, while **our method is completely training-free**. For this reason our main comparisons are restricted to other training-free methods. We will include an ablation study comparing to PDDS in the revision.
>
> > 3. The benefits of using SMC against MCMC (e.g. RDMC) with diffusion-based sampling methods are not clear.
>
> - **RDSMC serves as a correction mechanism to RDMC**. When using the same number of samples, discretization steps, and score estimation strategy, **RDSMC runs at the same cost of RDMC but offers asymptotic bias correction** (Theorem 1). In empirical study, we observe that **RDSMC consistently outperforms RDMC across targets**.
>
> - Moreover, **RDSMC provides unbiased estimates of normalization constants while RDMC and typical MCMC methods do not**.  This capability is useful for downstream tasks including computing free energy in statistical physics, Bayesian model comparison and density estimation.
>
> We will highlight these benefits in the revised paper.
>
> > 4. There are occasional typos that need to be fixed.
>
> Thanks for identifying these typos and we will fix them in the revision.
>
> > 5. It is claimed that the density $q(u_t \mid x_t)$ need not be evaluated (line 137). However, that seems to be needed to compute the importance weights (in Alg. 2).
>
> We thank the reviewer for pointing this confusion out. You are absolutely correct that the sampling distribution $q(u_t \mid x_t)$ is evaluated as part of the importance sampling-based score estimation procedure (Alg. 2). **What we intended to emphasize is that this distribution does not need to be evaluated in the outer SMC loop.** Specifically, $q(u_t \mid x_t)$ does not appear explicitly in the weight computation in Eq (15), as it cancels out between the proposal and target. We will revise the manuscript to clarify our intention.
>
>  >6. While in a different context, how is the design of the weighting functions similar to TDS?
>
>
> We thank the reviewer for pointing out this insightful connection between RDSMC and TDS (Wu et al 2023). While the goals and problem settings differ, **both methods incorporate “look-ahead” or "twisting" functions into the weighting functions.** These twisting functions introduce future information to intermediate states,  promote promising particles through the weighting procedure, and improve the SMC sampling efficiency.
>
> **In both TDS and RDSMC, the incremental weight expression takes the same form** (their Eq 11 v.s. Our Eq 15)
> $ \psi (x_t) \gamma (x_t, x_{t+1}) / \psi (x_{t+1}) q(x_t \mid x_{t+1})$ where $\psi$ denotes the twisting function, $\gamma$ is the untwisted target transition and $q$ is the proposal. While the exact definitions of these components are method-specific, this weight structure arises from the expression given optimal proposal and optimal twisting function (Naesseth et al, 2019, Chapter 3).
>
> Importantly,  this weight construction introduces twisting functions in a telescoping manner such that their approximation errors cancel out over the full trajectory. This property ensures that the final target is correct, and **the resulting SMC procedure is asymptotically exact for both TDS and RDSMC**.
>
> To further compare TDS and RDSMC, we clarify their exact settings:
> - TDS is designed to sample from the conditional $p_\theta (x_0 \mid y)\propto p_\theta(x_0) p(y\mid x_0)$ given a pretrained diffusion model $p_\theta(x_0)$ and a likelihood model $p(y\mid x_0)$ with observation $y$. **TDS’s twisting function** is $p(y \mid \hat x_\theta (x_t))$ (their Eq 8), **the likelihood evaluated at  the denoising prediction by the pretrained model** $\hat x_\theta (\cdot)$. This twisting function incorporates the observation $y$ into the weighting function (their Eq 11). Consequently, the particles that are better aligned with the observations $y$ are promoted.
>
> - In our case, RDSMC aims to sample from $\pi(x_0)$ using a diffusion-based proposal. We use $\hat p(x_t)$ (Eq 10) as **the twisting function, which approximates the exact marginal** $p(x_t) = \int p(x_t \mid x_0) \pi (x_0) dx_0$. This function integrates out the future information $x_0$, thereby encoding the target information $\pi(x_0)$ into the intermediate weight. The resulting weight expression (Eq 15) allows us to promote particles that are more consistent with the target $\pi(x_0)$.
>
> Finally, while both methods use twisting functions, **RDSMC introduces auxiliary randomness $u_t$ for estimating $\hat p(x_t)$, whereas TDS relies on a pretrained model**. This auxiliary randomness requires careful weight construction in RDSMC such that its influence cancels out properly (Eq 14 -> Eq 15). As a result, our method falls into the class of nested SMC methods, where TDS operates more in line with standard SMC.
>
> We will incorporate this discussion into the revised manuscript to clarify the similarities and distinctions between TDS and RDSMC.

---

> > ### Comment · Reviewer_8TQX · 2025-08-05
> >
> > I thank the authors for resolving my previous concerns. That said, the presentation of the main text needs to be clearer than is now, and I suggest the authors to highlight the contributions mentioned in the discussion in the revision. I will raise my score.

---

> > > ### Author Response · Authors · 2025-08-05
> > >
> > > We thank the reviewer again for their thoughtful feedback and we are glad to have addressed their concerns. We appreciate the suggestion to improve the clarity of the main text and to better highlight the contributions discussed. We will incorporate these changes in the revision to improve the overall presentation of the paper.

---

### Official Review · Reviewer_Kk4p · 2025-07-02

**Clarity:** 2
**Significance:** 3
**Originality:** 3
**Rating:** 4
**Confidence:** 1

**Summary:**

The authors developed a training-free diffusion sampler which treated diffusion-based sampler as the simulator within a sequential Monte Carlo framework. The authors proposed to resample the particles in each step to gradually steer the particles to the desired distributions and the issue of discretization bias and score estimation error can be greatly alleviated. The reverse diffusion SMC provides the unbiased estimate of the normalizing constant. Numerous methods show the promising potential of RDSMC compared to other diffusion samplers.

**Questions:**

1. what is the key step to avoid discretization error/ score error? is it Eq(15)? more clarifications are needed.

2. Is this algo. more like Wu's twisted SMC or vanilla SMC.

**Ethical Concerns:**

["NO or VERY MINOR ethics concerns only"]

**Final Justification:**

look-ahead mechanism seems to be helpful, but I am not fully convinced.

**Limitations:**

Although I appreciate the elegance, I am not convinced if this is a good idea to use SMC to correct the discretization bias and score error. Samplers are well-studied in low-dim distributions, but SMC doesn't scale well unless there is some clever tricks such as Tweedie-like "look-ahead" design.

**Quality:**

3

**Strengths And Weaknesses:**

Pros:

1) using SMC (Line 3 in Alg 2) to estimate score function via denoising score identity seems to be an interesting idea.

2) Eq(15) seems to have some potential alleviate the bias. I am not sure, need more evidence.

Cons:

1) I don't fully understand why the SMC setups can correct discretization bias and score estimation errors.

2) I don't understand the motivation to do training-free guidance in sampling setup, do we need a pretrained model in this case?

3) Give more details on the motivation and/or derivation of eq (6) and eq(14).


* I didn't carefully check the proof.

---

> ### Author Rebuttal · Authors · 2025-07-28
>
> We thank the reviewer for their careful and thoughtful evaluation. We are glad that they found our approach elegant and we appreciate their acknowledgement of the score estimation strategy and the potential of the bias correction mechanism. We find the questions and comments noted helpful and we address them below.
>
> Note: _if Latex equations do not render, please consider refreshing the page a couple of times_.
>
>
> > 1. Why can the SMC setups correct discretization bias and score estimation errors?
>
> The key reasons behind the correctness of RDSMC are: **(1) the final target distribution of RDSMC is explicitly constructed to match the desired target distribution, and (2) the score estimation errors only affect the proposals, which can be asymptotically corrected with SMC**. Our theorem 1 provides a rigorous statement with proof in the appendix.
>
> More specifically, Eq (13) shows that the final target distribution $\gamma_0(x_{0:T}, u_{0:T})$ is constructed such that its marginal $\gamma_0(x_0)$ exactly matches the desired target $\pi (x_0)$. This property holds regardless of the discretization scheme used, because **the desired target** $\pi (x_0)$ **is directly included in the final target** $\gamma_0 (x_{0:T}, u_{0:T})$, and when integrating out all other variables, only $\pi(x_0)$ remains, ensuring correctness of the marginal.
>
> The score estimation error also does not affect the correctness of the final target, as **the score estimates are only used in proposals**. As with any SMC method, **the resampling and reweighting steps (Eq 15) asymptotically correct for errors in proposals**, ensuring convergence to the final target in the limit of many particles.
>
> We will highlight these conceptual motivations and formal guarantees in the revised paper.
>
> > 2. What is the motivation to do training-free guidance in sampling setup, and do we need a pretrained model?
>
>  To clarify, **our method does not use training-free guidance, nor does it require a pretrained model**. Our method operates in a fully training-free setting, assuming only access to the unnormalized target density. We will make this point clearer in the revision to avoid confusion.
>
> > 3. Give more details on the motivation and/or derivation of eq (6) and eq(14).
>
> **Eq (6) computes the importance weights required for the score estimation, which reduces to the ratio of unnormalized target density and proposal density.**
>
> - By Tweedie’s formula or the Denoising Score Identity (Eq 3), computing the score is related to sampling from the posterior distribution $p(x_0 \mid x_t)$.
>
> - Since direct sampling from $p(x_0 \mid x_t)$ is intractable, we use importance sampling. Eq (6) computes the importance weights targeting $p(x_0 \mid x_t)$ using proposal $\mathcal{N}(x_0 \mid x_t / \alpha_t, \sigma_t^2 / \alpha_t^2)$.
>
> - Note that $p(x_0 \mid x_t) \propto \tilde \pi (x_0) N(x_t \mid \alpha_t x_0, \sigma_t^2I)$. Eq 6 computes the importance weights as the ratio of the unnormalized target density to the proposal density.
>
>
> **Eq (14) is a direct application of the standard SMC weighting procedure to our specific setting.**
>
> - As described in the background material (line 99), the incremental weight at each step is computed as the ratio of the current intermediate target and the product of the previous intermediate target and the proposal, i.e. $w_t \gets \gamma_t(x_{t:T}) / [\gamma_{t+1}(x_{t+1:T}) q(x_t \mid x_{t+1})]$.
>
> - By substituting our targets and proposals, we obtain the expression in Eq (14). Note that in our case, the targets and proposals are defined in the extended space which involve $u_{t:T}$.
>
>
> We will clarify the derivations and motivations of these equations in the revision.
>
> > 4.  I am not convinced if this is a good idea to use SMC to correct the discretization bias and score error. Samplers are well-studied in low-dim distributions, but SMC doesn't scale well unless there is some clever tricks such as Tweedie-like "look-ahead" design.
>
> We appreciate the reviewer’s point about the importance of look-ahead designs in high dimensions. In fact, **our approach does incorporate a look-ahead mechanism through the marginal density** $p(x_t)=\int p(x_t \mid x_0) \pi(x_0) dx_0$​, which integrates the future information $\pi(x_0)$.
>
> These marginals serve as the foundation for defining our optimal intermediate target distributions (see Eq. 9), akin to the goals of twisting/ look-ahead adjustments in other SMC literature (e.g. TDS in Wu et al 2023). When the exact marginals are known, one can in principle sample accurately even in high dimension.
>
> In practice, we estimate the marginal $p(x_t)$ using approximations (Eq. 10) and derive tractable "twisted" intermediate targets (Eq 11) and weights (Eq 15). These marginal approximations are increasingly accurate as the sampling trajectory progresses, and provide effective look-ahead guidance to improve sampling efficiency.
>
> In empirical study, RDSMC consistently outperforms _RDSMC (IS)_ (which does not provide "look-ahead" mechanism in intermediate states and only performs IS in the end), and _RDSMC (Proposal)_ (which does not involve any importance weighting correction). **This comparison highlights the effectiveness of the look-ahead mechanism and SMC correction.**
>
> We will clarify the connection to look-ahead SMC techniques in the revised paper.
>
>
> > 5. What is the key step to avoid discretization error/ score error? is it Eq(15)? more clarifications are needed.
>
> Please see above response to the first question.
>
>
> > 6. Is this algo. more like Wu's twisted SMC or vanilla SMC.
>
> Structurally, our method shares similarities with Twisted Diffusion Sampler (TDS) (Wu et al 2023), as **both methods use some form of "look-ahead"  or "twisting" functions to improve the efficiency of SMC.**  These look-ahead functions introduce future information to intermediate states,  promote promising particles through the weighting procedure, and improve the SMC sampling efficiency.
>
> More concretely, **the incremental weight expression of both TDS and RDSMC takes the same form** (their Eq 11 v.s. Our Eq 15)
> $ \psi (x_t) \gamma (x_t, x_{t+1}) / \psi (x_{t+1}) q(x_t \mid x_{t+1})$ where $\psi$ denotes the look-ahead/twisting function, $\gamma$ is the untwisted target transition and $q$ is the proposal. While the exact definitions of these components are method-specific, this weight structure arises from the expression given optimal proposal and optimal twisting function (Naesseth et al, 2019, Chapter 3).
>
> Importantly,  this weight construction introduces twisting functions in a telescoping manner such that their approximation errors cancel out over the full trajectory. This property ensures that the final target is correct, and **the resulting SMC procedure is asymptotically exact for both TDS and RDSMC**.
>
> However, we want to highlight **several important differences between TDS and RDSMC**:
>
> (1) **Problem setting:** The problem we address is fundamentally different from that of TDS. While TDS is designed for sampling from a distribution tilted by a pretrained diffusion model, our goal is to sample from an arbitrary distribution, assuming access to its unnormalized density.
>
> (2) **Design of key components:** Due to the difference in settings, our target distributions, proposal distributions, and the look-ahead functions require significantly different designs. In particular,
>   -  TDS is designed to sample from the conditional $p_\theta (x_0 \mid y)\propto p_\theta(x_0) p(y\mid x_0)$ given a pretrained diffusion model $p_\theta(x_0)$ and a likelihood model $p(y\mid x_0)$ with observation $y$. **TDS’s twisting function** is $p(y \mid \hat x_\theta (x_t))$ (their Eq 8), **the likelihood evaluated at  the denoising prediction by the pretrained model** $\hat x_\theta (\cdot)$. This twisting function incorporates the observation $y$ into the weighting function (their Eq 11). Consequently, the particles that are better aligned with the observations $y$ are promoted.
>
>   - In our case, RDSMC aims to sample from $\pi(x_0)$ using a diffusion-based proposal. We use $\hat p(x_t)$ (Eq 10) as **the twisting function, which approximates the exact marginal** $p(x_t) = \int p(x_t \mid x_0) \pi (x_0) dx_0$. This function integrates out the future information $x_0$, thereby encoding the target information $\pi(x_0)$ into the intermediate weight. The resulting weight expression (Eq 15) allows us to promote particles that are more consistent with the target $\pi(x_0)$.
>
>
> (3) **Use of auxiliary variables:** Our method introduces auxiliary randomness through the Monte Carlo estimation of the look-ahead function. Incorporating this randomness within our framework places our approach in the category of nested SMC methods (Naesseth et al 2015). In contrast, TDS does not use auxiliary variables and operates more like a standard SMC algorithm, without requiring nested sampling.
>
> We will clarify the connections and differences in the revised paper.

---

> > ### Author Response · Authors · 2025-08-06
> >
> > Dear reviewer,
> >
> > As the discussion period is nearing its end, we wanted to kindly follow up to see if there are any remaining questions we can address. Since you had raised important points earlier which we have in turn addressed, your updated feedback would be helpful to ensure the paper is fairly assessed. We would be glad to respond to any further questions you may have.
> >
> > Thank you again for your time and thoughtful review.

---

> > ### Comment · Reviewer_Kk4p · 2025-08-06
> >
> > Thanks for the feedback. I have read the rebuttal and slightly improved the score with the lowest confidence, but I have to acknowledge that I don't fully understand the work. I will leave the judgment to other reviewers.

---

> > > ### Author Response · Authors · 2025-08-07
> > >
> > > We thank the reviewer for taking the time to read our rebuttal and for improving the score. We appreciate their engagement and openness in deferring to others for the final judgement. The discussion has been helpful for improving the paper, and we will incorporate these points into the revision.

---

### Official Review · Reviewer_5DzY · 2025-07-03

**Clarity:** 3
**Significance:** 2
**Originality:** 2
**Rating:** 4
**Confidence:** 2

**Summary:**

- The paper leverage the diffusion model paradigm to sample from a target distribution, which can be unnormalized.
- It introduces a training-free Sequential Monte Carlo (SMC) approach where the proposal are derived from a reverse diffusion kernel.
- The core idea is to augment the state space with an auxiliary variable that accounts for the randomness in the score estimation.

**Questions:**

.

**Ethical Concerns:**

["NO or VERY MINOR ethics concerns only"]

**Final Justification:**

- The proposed method is novel
- Following the authors' clarification regarding my concern about the high-variance of the proposal, I have a better view the existing methods
- I am now updating my rating to reflect this new understanding and the strength of the work

**Limitations:**

.

**Quality:**

1

**Strengths And Weaknesses:**

## Strength

- The main novelty of the paper is the idea of augmenting the state space with an auxiliary variable to explicitly model the noise in the score estimation process.

## Weaknesses

**Major Theoretical Flaw**

The central methodology for score estimation appears to be fundamentally flawed:

- The proposed score estimation in Equation (5) is problematic. By examining the weight update in Equation (6), it appears the auxiliary variable $u$ is intended to act as a substitute for $x_0$ within the target density term.
- However, $u$ is sampled from the proposal distribution $N(x_t / \alpha_t, \sigma_t^2 / \alpha_t^2 I)$, which is a Gaussian with a very large variance, particularly in early steps. This means that samples of $u$ have a low to no chance of falling in regions where the target distribution has high mass.
- This turns the score calculation into an IS estimate where the proposal has a poor overlap with the target. Consequently, estimating the score in this way is likely as difficult as sampling from the target distribution directly, which defeats the purpose of the proposed method.

**Typos**

- In Line 96: "initilaizes" should be corrected to "initializes".
- There is a trailing comma in the denominator of the weight expression in Equation (6) that should be removed.

---

> ### Author Rebuttal · Authors · 2025-07-28
>
> We thank the reviewer for their time and effort in evaluating our work, and we appreciate their acknowledgment of the novelty of our work in augmenting the state space to explicitly model the noise in the score estimation process.
>
> However, we respectfully disagree with the reviewer's assessment of the score estimation strategy, as it overlooks established prior work and does not fully capture the intended flexibility and central designs of our method. We outline these points below and encourage the reviewer to reconsider the assessment in light of the following clarifications:
>
> - **Eq (5) is built on established methodology:** the IS score estimator (Eq 5) is grounded in a well-established line of works which reverse the forward transition kernel to obtain the proposal $\mathcal{N}(x_0 \mid x_t / \alpha_t, \sigma_t^2 /\alpha_t^2)$ to estimate scores (e.g. Huang et al 2023, Akhound-Sadegh et al 2024). While the proposal’s variance is large in earlier steps, it decreases to 0 as $t \to 0$ and the proposal becomes more informative, making the score estimates more accurate. More concretely,
>
>   - Huang et al 2023 consider a similar importance weighted score estimator (see their Eq 5 in Section 3.3). Their Theorem 1 shows that  sampling from the posterior $p(x_0 \mid x_t)$ is easier than sampling from the original target $\pi(x_0)$, as indicated by the improved LSI constant. Empirically, they observed that "When $t$ is close to 0, we are able to quickly obtain rough score estimates via the importance sampling approach".
>
>    - Akhound-Sadegh et al 2024 use $\mathcal{N}(x_0 \mid x_t, \sigma_t^2)$ as the proposal distribution to form a self-normalizing estimate (see their Eq 8), based on the Target Score Identity (our Eq 19 in appendix). This proposal is equivalent to ours, as $\alpha_t$ is set to 1 in their Variance-Exploding diffusion schedule. They also provide bounds on this score estimator.
>
>
> - **Illustrative role of Eq 5 and flexibility of our approach:** As stated in the paper (line 129), the estimator in Eq (5) is meant as an illustrative example and our framework can accommodate different score estimators. We discuss more advanced score estimators in appendix B.1, as suggested in line 133-134 below Eq (5) and (6).  These strategies include annealed importance sampling (Alg. 3 in appendix), more informed IS proposals (appendix B.1.3.) and estimators based on Target Score Identity (Eq 19 in appendix) or Mixed Score Identity (Eq 20 in appendix).
>
> - **Addressing proposal bias is a central design of our work:** The review focuses on the potential bias of a particular score estimator used to construct the proposal, but our method is explicitly designed to account for such bias. A key strength of RDSMC is its ability to incorporate biased proposals within the SMC algorithm and progressively correct for these biases. Theorem 1 provides theoretical guarantees. We also illustrate this point in empirical study. For example, as shown in Figure 1a , using the proposal alone yields biased results, while RDSMC reduces this bias.
>
> We will clarify these points in the revision to prevent potential misunderstandings and to better highlight the flexibility and theoretical guarantees of our framework.
>
> ---
>
> We also thank the reviewer for identifying the typos and we will fix them in the revision.
>
> Note: _if Latex equations do not render, please consider refreshing the page a couple of times_.

---

> > ### Comment · Reviewer_5DzY · 2025-08-04
> > **Rely**
> >
> > I thank the authors for their response and acknowledge reading their reply.
> >
> > The proposal $N(x_t / \alpha_t, \sigma_t^2 / \alpha_t^2 I)$, while having high variance, has indeed been adopted in several notable works. I appreciate the authors' clarification on this point and will update my assessment of the work accordingly.

---

> > > ### Author Response · Authors · 2025-08-04
> > >
> > > Thank you for your reply. We are glad to have addressed your concerns.

---

> > > > ### Author Response · Authors · 2025-08-07
> > > >
> > > > Dear reviewer,
> > > >
> > > > Thank you again for your engagement during the discussion period, and we are glad to have resolved you previous concern regarding a specific proposal choice.
> > > >
> > > > As the discussion window is nearing its end, we wanted to kindly follow up in case there are any remaining questions related to your reassessment of the paper. We would be happy to clarify anything further.

---

### Official Review · Reviewer_oZor · 2025-07-04

**Clarity:** 3
**Significance:** 2
**Originality:** 3
**Rating:** 4
**Confidence:** 3

**Summary:**

This paper combines the sequential Monte Carlo with the reverse diffusion path to provide a novel sampler to draw samples from an irregular target distribution. Different from typical sequential Monte Carlo, this paper introduces auxiliary variables $u_t$ to construct the proposal distribution $q(x_t|x_{t+1},u_{t+1})$ for simulating the transition in the reverse diffusion process and calculating the weights for the resampling. Different from RDMC, this paper will keep a sequence of particles in each update and use them to construct the proposal distribution for resampling. As a result, this paper shows that sequential Monte Carlo will be an unbiased sampler when the number of particles in the sequence approaches infinity and empirically shows the advantages of their methods.

**Questions:**

Please refer to the weaknesses part.

**Ethical Concerns:**

["NO or VERY MINOR ethics concerns only"]

**Final Justification:**

In the rebuttal period, the authors have addressed most of my questions on the theory, and therefore, I feel comfortable raising my score.

**Quality:**

2

**Strengths And Weaknesses:**

**Strength:**

1. The problem expected to be solved (simulating reverse diffusion path to solve the sampling task) is interesting, and the core technique sequential Monte Carlo helps to approximate the ideal transition kernel $p_{0|t}$ by re-weighting the proposal with $w_i$ is practical since it has bypassed repetitive posterior sampling in previous work, e.g., RDMC.
2. The core technique, i.e., introducing the auxiliary variables $u_t$ to couple sequential Monte Carlo and RDMC is original and effective, which helps to provide unbiased score estimation in expectation. Although the unbiasedness is conditioned on the infinite particle, it still makes a difference compared with RDMC.
3. This paper is well written and makes it easy for readers to follow.

**Weaknesses:**

1. The theoretical results provided in this paper are limited. Some important theoretical properties for the proposed algorithm are unknown. For example, can the analysis be extended to the finite particle $N$ case? Will the algorithm converge? What kinds of convergence (high probability or in expectation) will be achieved? What is the size of particles required to be preserved to guarantee the convergence? Since the rejection sampling in high dimensions will bring the dimension curse, while this curse is preserved in the sequential Monte Carlo?
2. The significance of the advantages of reverse diffusion SMC is limited. In the 2-d case, the improvement of TV distance is limited. Under the 10-d case, neither sliced KSD nor logZ estimation will have advantages. Although the algorithm is elegant and insightful, its effectiveness may be limited.
3. Authors may add an independent table to clarify the notation, which will improve the readability of this paper. Currently, the notations are usually hidden in different parts of the paper.

---

> ### Author Rebuttal · Authors · 2025-07-28
>
> We thank the reviewer for their detailed review, and we are glad that they found the paper well-written and the proposed algorithm elegant and insightful. We find the questions and suggestions helpful and we address them in response below.
>
> Note: _if Latex equations do not render, please consider refreshing the page a couple of times_.
>
>
> > 1. The theoretical results provided in this paper are limited.
>
> Our method can be set up as a standard SMC on an extended space (similar to constructions such as nested SMC (Naesseth et al 2015)). As a result, **theoretical properties of SMC apply directly to our method**:
>
> - **Finite particle guarantee**: for any particle size $N\geq 1$, any test function $\phi()$ integrating with respect to the unnormalized target can be estimated unbiasedly (e.g. Naesseth et al, 2019, Theorem 2.3.1). We obtain the unbiased estimate of the normalization constant for $\phi()=1$.
>
> - **Asymptotic convergence**: as $N\to \infty$, under standard regularity assumptions, SMC estimates converge to the true expectation in probability 1 (e.g. Chopin and Papaspiliopoulos, 2020, Proposition 11.4, or Theorem 3 restated in our appendix). Under regularity conditions, the central limit theorem holds for SMC with the same rate ($1/\sqrt{N}$) as MCMC (Chopin 2004).
>
> - **Curse of dimensionality**: SMC scales well for discretization steps $T$ (usually $N$ needs to scale as $T$ for marginals to be stable, see Proposition 11.13 in Chopin & Papaspiliopoulos (2020) ), but just like other importance sampling methods it requires careful proposal/target choice to scale with dimension $d$ of the latent state. Chatterjee and Diaconis (2018) show that for stability $N$ needs to scale with $\exp(\textrm{KL}(p||q))$ where $p$ is the target and $q$ the proposal. In practice, for well-chosen targets and proposals we can still get well-performing algorithms for high-dimensional $x_t$ (e.g. Naesseth et al, 2019, Chapter 3).
>
> We will include this discussion in the revision to clarify the theoretical properties of our method and their foundations in the classic SMC literature.
>
> Additional reference:
> _Nicolas Chopin. Central Limit Theorem for Sequential Monte Carlo Methods and Its Application to Bayesian inference. Annals of Stats, 2004._
>
>
> > 2. The significance of the advantages of reverse diffusion SMC is limited.
>
> We respectfully disagree with the reviewer’s assessment that the effectiveness of RDSMC is limited. In fact, we believe our results demonstrate that RDSMC offers meaningful advantages as it uses fewer discretization steps and does not require access to target information. Moreover, RDSMC extends existing diffusion-based methods by enabling log normalization constant estimates. Below we clarify key points and provide supporting ablation results:
>
> - **RDSMC shows better or competitive performance without access to target variance and with fewer discretization steps:** As described in line 249-255, competing methods including **AIS, SMC and SLIPS have access to the target’s scalar variance and use substantially more discretization steps** (T=1,024); SLIPS also uses an SNR-adapted time discretization schedule and a Langevin-based initialization. In contrast, RDSMC is run with $T=100$ uniformly spaced steps, does not use target variance and starts from a simple Gaussian $N(0,1)$. Despite these differences, RDSMC matches or outperforms these methods in most cases. In particular, RDSMC outperforms SLIPS along other baselines in the GMM weight ratio estimation task across varying dimensions $d \in [2,64]$  (Figure 2 in appendix).
>
> - As a more neutral comparison, **RDMC is similar to SLIPS but  does not assume target variance, and RDSMC consistently outperforms RDMC across multiple targets**.
>
> - **RDSMC outperforms baselines given target variance**: To further isolate the effect of our core contributions, we include an ablation experiment on rings and funnel targets (see table below), where all methods use the target scalar variance (RDSMC uses it to initialize the proposal for score estimation). In these settings, RDSMC outperforms SMC and SLIPS on most metrics.
>
>
> | Method                            | Rings TVD               | Rings Log Z bias        | Funnel KS               | Funnel Log Z bias        |
> |----------------------------------|--------------------------|--------------------------|--------------------------|---------------------------|
> | RDSMC (T=200, use target var)    | $\mathbf{0.102 \pm 0.002}$ | $\mathbf{0.012 \pm 0.004}$ | $\mathbf{0.065 \pm 0.002}$ | $\mathbf{0.219 \pm 0.025}$ |
> | SMC (T=1024, use target var)     | $\mathbf{0.096 \pm 0.002}$ | $0.053 \pm 0.002$         | $0.071 \pm 0.000$         | $0.283 \pm 0.005$          |
> | SLIPS (T=1024, use target var)   | $0.189 \pm 0.003$         | NA                       | $\mathbf{0.063 \pm 0.001}$ | NA                        |
>
>
>
> - **Additional capability to estimate normalization constants $Z$**: Unlike SLIPS and RDMC, RDSMC provides $\log Z$ estimates **without additional compute cost** (given the same number of generated samples, discretization steps and same score estimation strategy). This estimate is useful for downstream tasks including computing free energy in statistical physics, Bayesian model comparison and density estimation.
>
> We will revise the manuscript to clarify these distinctions in settings and include ablation study results.
>
> > 3. Authors may add an independent table to clarify the notation, which will improve the readability of this paper. Currently, the notations are usually hidden in different parts of the paper.
>
> We thank the reviewer for the suggestion and we will clarify notations in the revised version.

---

> > ### Author Response · Authors · 2025-08-06
> >
> > Dear reviewer,
> >
> > As the discussion period is nearing its end, we wanted to kindly follow up to see if there are any remaining questions we can address. Since you had raised important points earlier which we have in turn addressed, your updated feedback would be helpful to ensure the paper is fairly assessed. We would be glad to respond to any further questions you may have.
> >
> > Thank you again for your time and thoughtful review.

---

> > ### Comment · Reviewer_oZor · 2025-08-07
> >
> > The authors have addressed most of my questions on the theory, and therefore, I feel comfortable raising my score. I encourage the authors to further explore and clarify their potential application domains in future work.

---

> > > ### Author Response · Authors · 2025-08-07
> > >
> > > We thank the reviewer for their reply and we are glad to have addressed the reviewer's concern.
> > >
> > > We appreciate the reviewer's suggestion to explore potential application domains. One exciting direction is to combine our current framework with pre-trained diffusion samplers as proposal distributions, leveraging their learned representations while correcting their biases in a principled way. This ability could enable scalable amortized sampling for larger or diverse biophysical systems .We will clarify this direction in the revision.

---

### Note · Authors · 2025-08-14

We thank all the reviewers for their engaging discussion. They recognized the **novelty** of using SMC in an augmented diffusion state space to approximate optimal transitions (oZor, 5DzY, 8TQX), and the **theoretical guarantees** for consistent sampling and unbiased estimation of the normalization constant (oZor, 8TQX).

We are glad that our responses have addressed their concerns and led to improved assessments. Below we summarize the key points and our responses.

- **Theoretical properties (oZor):** Our method is an SMC algorithm defined over the extended state space of reverse diffusion dynamics. Therefore it inherits all standard SMC guarantees, including finite-particle bounds, asymptotic convergence, and scaling behaviors in high dimensions.
- **Empirical advantages (oZor, 8TQX):** Our method outperforms or matches baselines with fewer discretization steps and without using target variance. We provide additional results where our method also uses target variance which outperforms baselines. Moreover, our method provides unbiased estimate of normalization constant while prior training-free diffusion baselines do not.
- **Score estimator in Eq 5 (5DzY):** This estimator is built upon established prior work and serves as  an illustrative choice within our general framework. We also evaluate other score estimators in the paper, and the key strength of our approach is to incorporate any score estimator in the proposal and mitigate their biases with SMC.
- **How SMC corrects discretization bias and score estimation bias (Kk4p):** Our theorem 1 provides formal guarantees. The reasoning is (1) the final target in SMC is explicitly constructed to match the desired distribution which is agnostic to the discretization, and (2) the score estimation errors only affect the proposals and can be asymptotically corrected by SMC.
- **Connection to TDS (Wu et al., 2023) (Kk4p, 8TQX):** While both methods use SMC and “twisting” or “look-ahead” functions, they address different sampling problems. TDS targets tilted distribution of a pretrained diffusion model, whereas we target any unnormalized analytic distribution. This difference requires different designs for the key SMC components, including the proposal mechanism, weighting scheme, and use of auxiliary variables.

In the revision, we will clarify the above points, add intuitions for key technical ideas, include ablation results in the discussion, and incorporate other writing suggestions from the reviewers.

---

### Decision · Program_Chairs · 2025-09-17

**Decision:**

Accept (poster)

**Comment:**

The motivation of the work is the weakness of current diffusion-based samplers for unnormalized targets. Reverse diffusion methods rely on approximate scores and discretization, which introduce bias, and existing Monte Carlo corrections remain inefficient. The paper proposes Sequential Monte Carlo (SMC) as a natural framework to correct these issues, by reweighting and resampling biased diffusion-based proposals.

The main contribution is the design of a reverse diffusion sequential Monte Carlo algorithm. The method augments the state space with auxiliary variables that capture randomness in score estimation and incorporates them into the proposal and weights. This allows diffusion-based proposals to be used while their bias is corrected by SMC. The algorithm inherits standard and well-studied SMC guarantees and enables unbiased estimation of normalization constants, something not offered by competing samplers. Empirically, the method is competitive with or stronger than baselines, especially when fewer steps or no target variance are assumed.

The discussion clarified open questions. Theoretical guarantees were confirmed to follow from standard SMC theory on an extended space. Empirical concerns were addressed by pointing out that competing methods often use more computation or extra information, while ablations showed competitive or superior performance under matched conditions. The relation to twisted SMC was explained, with similarities in weight structure but differences in setting, design, and use of auxiliary randomness. The authors also committed to improve clarity and notation. Most reviewers raised their scores after these clarifications.

In conclusion, the paper provides a principled and original integration of diffusion-based proposals with sequential Monte Carlo, supported by theory and empirical results. Despite some clarity issues, the contribution is sound and of value to the ML / Monte-Carlo community.